# Monitoring Sea Currents with Midrange Acoustic Backscattering

Alexey V. Ermoshkin [1], Ivan A. Kapustin [1], Dmitry A. Kosteev [1], Alexander A. Ponomarenko [1,2,*],
Dmitrii D. Razumov [1,*] and Mikhail B. Salin [1,*]

1   Center for Hydroacoustics and Geophysical Research Division, Institute of Applied Physics of the Russian
    Academy of Sciences, 603950 Nizhny Novgorod, Russia; eav@ipfran.ru (A.V.E.); kia@ipfran.ru (I.A.K.);
    dkosteev@ipfran.ru (D.A.K.)
2   Laboratory of Algorithm and Technologies for Network Analysis, HSE University,
    603036 Nizhny Novgorod, Russia
*   Correspondence: aponomarenko@hse.ru (A.A.P.); ddrazumov@ipfran.ru (D.D.R.); mikesalin@ipfran.ru (M.B.S.)

**Abstract:** This paper is devoted to an acoustical method of measuring mesoscale sea and ocean currents. Due to the fact that such currents exhibit variability, long-term studies are of great interest. The aim of this study is to prepare a physical foundation to organize current measurements in an automated way using stationary mounted underwater echosounding systems. An acoustic system operating at a frequency of 1–3 kHz (lower than commercial frequencies) that is capable of sensing echo signals from natural inhomogeneities located at distances of 1 to 10 km was tested. The test was conducted during a two-week marine experiment on the northern shelf of the Black Sea. The acoustic system was mounted on a platform together with a weather station and other tools that provided reference values for further comparison. Scattering from moving particles, as well as from wind waves, provides a general opportunity for sensing of currents at remote points. Since most scatterers exist at a depth of at least 2 m or on the surface, the proposed sensing method is going specialized for currents in upper layers. However, analysis of Doppler spectra of the actual returning (reverberation) signal showed that this kind of scattering was mixed with bottom reverberation (which contains no additional frequency shift), and other signal distortions were present. Thus, we proposed a new method of signal processing that is aware of the regional environment. The described method is based on machine learning, namely on gradient boosting to build decision trees, which compute water current properties. Such a computational routine is preceded by an original acoustic signal feature extraction process. Finally, a precision of an order of magnitude was achieved, and a sensing distance of at least 2 km was proven as a result of this study carried out with available instruments.

**Keywords:** low-frequency reverberation in the sea; backscattering; gas bubbles in water; wind waves; acoustic reverberation spectrum; machine learning; upper-layer current





## 1. Introduction

Sea and ocean currents have a profound effect on marine life, not only moving animals and plants across the ocean but also redistributing heat and nutrients. In addition, ocean currents have an impact on human activities, such as ship navigation, operations from platforms, etc. Currents are detected in a wide range of scales. For example, in the open ocean, currents may move around small sub-mesoscale features only a few hundred meters in size or around mesoscale features a few tens of kilometers across, such as ocean rings and eddies, or may flow across or around entire ocean basins, including well-known features such as the Gulf Stream (North Atlantic), the Kuroshio Current (North Pacific), etc. Major ocean currents are well-described, since global observation of ocean currents is essential for problems of heat exchange and weather and climate forecasting. Regional currents, namely small-scale currents in harbors and straits, might be affected by many factors, such as wind direction and time of year. Measuring coastal currents is a topical problem, since it

affects shipping; the drift of hazardous objects, such as sea mines and water pollutants; the erosion of natural beaches; and changes in coastlines [1].

There are many ways to measure/estimate the current velocity in the upper layer of a water area [2]. Local probes are convenient, considering an array of contact sensors located in the water [3], e.g., stationary sensors (propeller, electromagnetic or acoustic) or drifting sensors, tracking their position [4,5]. An acoustic Doppler current profiler (ADCP) uses four transmitters and receivers of high-frequency sound of 75 kHz–2.4 MHz, the beams of which are directed close to the vertical column [6,7]. An onboard computer analyzes delays and Doppler frequency shifts of the signal and can provide information on the current velocity distribution in the vertical water column. However, these methods might be expensive for large-scale problems.

Remote sensing is a promising approach. Radar methods for monitoring sea currents [8], as well as some optical methods, are based on the fact that surface waves scattering radiation are driven by the current. In simple terms, the current velocity in the upper layer can be estimated from the Doppler shift of the scattered signal, which corresponds to the sum of the traveling wave-phase velocity and the surface current velocity. In fact, the scattered signal is characterized by a certain spectrum that requires proper statistical processing. Such data are available if coherent radar is used. In the case of incoherent methods, one can observe the wave structure and evaluate the change according to the dispersion relation.

Underwater sound that passes through the water column makes it possible to obtain an integral estimation of the current velocity in the projection on the beam. Obtaining such integral estimations at long distances using horizontal beams is of great interest for a number of practical problems. For example, some works describe a network of receivers and sources for acoustic tomography that operate at a frequency of about 5 kHz [9–11]. This network can operate on opposite shores of narrow straits with a width of 1–2 km and collect information on the flow by analyzing the variations in propagation time of sound pulses between two or more control points.

This paper is devoted to a method that exploits the Doppler effect in horizontal backscattering to assess the parameters of currents within an area of several kilometers. An experiment with an underwater acoustical system, the operational range of which, in terms of sensing currents, turned out to be approximately 2 km, is described in this paper. We plan to extend the proposed system to reach a distance of 10 km in future research. Long operational distances can be achieved due to the low frequency of the transmitted signals, i.e., several tones in the 1–3 kHz band, which is much lower than the frequency of commercially available ADCPs. In contrast to acoustic tomography systems, this system exploits a collocated transmitter and receiver; therefore, it can be used in coastal zones facing the open sea. This work can be regarded as a small step in this direction.

In Section 2, we provide some initial information that led to the chosen system design. Section 3 describes instruments, the scheme of the experiment, and some common signal processing routines. In Section 4, we propose a machine-learning-based method of signal processing and current velocity estimation. In Section 5, we provide a correlation analysis of the experimental data. In Section 6, we discuss the origin of errors, compare the achieved parameters of the proposed method with those of other methods, and propose some improvements for future studies.

## 2. Preliminary Analysis

When designing a system capable of echoing currents over several kilometers, we chose a frequency range of 1–3 kHz because of its low sound attenuation in water, as previously noted [12]. The intensity of our underwater transducers is sufficient to ensure that the signal reflected from natural inhomogeneities over several kilometers is significantly stronger than any noise interference upon returning to the sonar frequency.

For comparison, we looked at similar systems, such as CODAR-type radars [13–15], which are sensitive to three-meter-long surface waves, and horizontally positioned ADCPs

that scatter radiation on small inhomogeneities in the medium [6]. In our case, as sound propagates over long distances in the wave guide between the surface and the bottom, it is scattered by both surface waves and medium inhomogeneities. However, it is important to note that the scattering characteristics of inhomogeneities in the medium at the studied frequency range differ from typical values in the frequency range of ADCP operation. This is because the resonant length of the surface wave is 36 cm for sound with a frequency of 2 kHz (Acoustic Bragg backscattering on surface waves takes place when the surface wavelength matches half of the sound wavelength. Taking into account that the average value of the sound speed in water is 1450 m/s, one may figure out that the sound wavelength is 72.5 cm at 2 kHz and that the rounded value of the corresponding surface wavelength is 36 cm). Therefore, our proposed transition to the middle range of sound frequencies requires separate experimental validation.

To achieve the required Doppler frequency resolution, long-tone pulses containing about a thousand oscillation periods are used. When one transmits a $\delta t$-duration pulse and performs a Fourier transform with a matching window size, a frequency resolution of $\delta f = 1/\delta t$ must be achieved. On the basis of this fact, the following expression for Doppler velocity resolution ($\delta v$) can be derived:

$$\delta r \delta v = \frac{1}{4} \lambda c \qquad (1)$$

Regarding cell size, $\delta r$ is determined by the sound wavelength ($\lambda$), sound speed ($c$), and flow velocity. For industrial ADCPs operating at centimeter wavelengths, $\delta r$ should be about 37.5 m when measuring flows at a speed of 0.1 m/s. In contrast, the cell size of our proposed system is around 2.6 km for the same current velocity. However, Equation (1) is valid when applying Fourier transform to the backscattered signal or analogous techniques. High-frequency (HF) radars use chirp signals but still suffer from low-range resolution [13], with $\delta r$ values ranging from 0.25 to 7.5 km depending on the chirp bandwidth, which is 600 kHz and 20 kHz, respectively.

We chose long-tone pulses for their narrow spectrum, which allows us to study the distribution of signal reverberation over Doppler frequencies in detail. However, the disadvantage is a large scattering zone. Continuous-wave (CW) signals are not preferable, since pulses allow us to choose when reverberation occurs after the pulse. Researchers have also reported that pulses are more pleasant to hear than CW transmission, since we are working with sound in the audible range.

The sound is scattered in the opposite direction by the bottom, water column, and surface. The contribution of the bottom to the change in the Doppler spectrum of the scattered signal is considered insignificant compared to the emitted signal, but the bottom scattering contains a large amount of energy, making the "center of mass" of the spectral distribution less sensitive to changes in other parts of the spectrum.

In previous research, we constructed a machine learning model to predict surface wave height, period, and wave vector based on midrange sound backscatter spectrum features [16]. In this study, we use data collected from the same experiment and using the same methodology but employ current upper-layer (1–2 m depth) data measured by ADCP as true values to train the model rather than buoy wave data. To achieve this, we slightly modify acoustic signal processing by extracting reverb with minimal delay and reducing the number of pulses for averaging.

## 3. Materials and Methods

The experiment on remote sensing and measurement of the sea state were conducted on an oceanographic platform in the Black Sea for two weeks in the second half of September 2021. Note that other remote sensing devices, namely x-band radar [17] and stereo photography [18], were used simultaneously with underwater acoustics; thus, hope to present combined processing of all types of data elsewhere. An industrial acoustic Doppler current profiler (ADCP WorkHorse Monitor 1200 kHz RDI TeledyneTM) was located near

the surface (when looking down). Despite the fact that it provided the velocity profile at different horizons, the actual values in the upper (1–2 m) and lower (about 10 m) layers appeared to be approximately equal in this experiment.

We use a sound emitter and an array of hydrophones produced by our institute subdivision. A ring-type omnidirectional ceramic emitter produced an acoustic pressure of 1 kPa at a distance of 1 m while driven by a 3 kW power amplifier of the Cerwin-Vega brand, followed by a choke to balance the electric loading. The antenna initially contained 32 hydrophones, but in the experiment, the contact was poor. and only 13 consecutive hydrophones were used. The step between hydrophones is 0.2 m. The frequency range of a single hydrophone is 5–12,000 Hz. The antenna is usually employed in the range of 200–3500 Hz, where two criteria are met: the antenna length should be more than a wavelength, with less than a half wavelength spacing between the neighbor hydrophones. Each sensor has a flat response in that band and a sensitivity of around 5 mV/Pa. Less than 3 dB difference in sensitivities of all sensors was provided.

The scheme of the low-frequency acoustic experiment with the arrangement of devices on the platform is shown in Figure 1a. The horizontal line of hydrophones and a transducer (sound emitter) were deployed from the platform (suspended on ropes) to depths of 13 m and 18 m, respectively. The depth to the bottom near the platform was 30 m. The emitted signal was a rectangular pulse, with a duration of 2 s associated with carrier frequencies of 1320, 2020, and 2720 Hz and a duration of 8 s associated with a frequency of 2080 Hz. The pulse repetition period was 90 s.

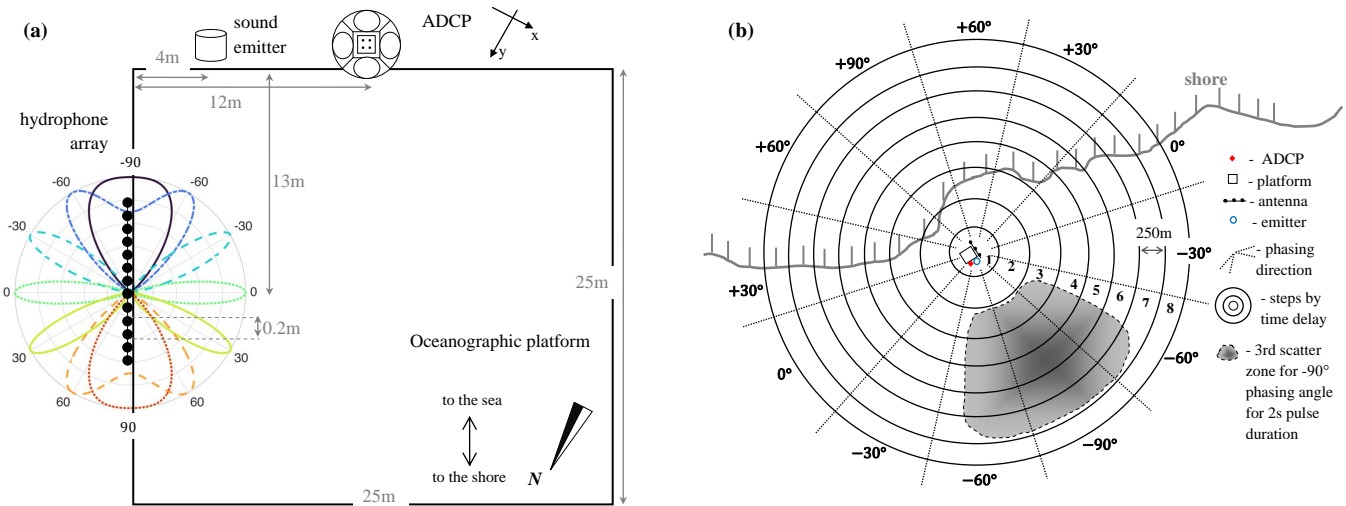

**Figure 1.** Scheme of the experiment in the horizontal plane: (**a**) the platform with sensors and antenna directional pattern; (**b**) map with the platform in the center.

The original signal ($p_n(t)$) from $n = 1 \ldots 13$ hydrophones recorded at a sampling frequency of 25 kHz was converted to reduce the amount of data fed to the input of the model discussed below. The real-valued signal ($p_n(t)$), which is concentrated near carrier frequencies of $f_0$, is related to its complex envelope ($C_n(t)$) via the following expression:

$$p_n(t) = C_n(t) \exp(-i2\pi f_0 t) \tag{2}$$

where $C_n(t)$ is extracted from $p_n(t)$ using a conventional digital signal processing routine, i.e., heterodyning, which is performed for each four carrier frequencies ($f_0$). The resulting $C_n(t)$ values are stored in files. Such a signal has a reduced sampling rate of up to 50 Hz. The next step is phasing (3), which is performed via weighted summation of the complex

signal from the hydrophones with coefficients that depend on the phasing direction ($\theta$) and the hydrophone position.

$$S(\theta, t) = \sum_{n=1}^{13} \frac{\text{hanning}(n,13)}{13} \exp\left(i\sin(\theta)\frac{2\pi f_0}{c} x_n\right) C_n(t) \qquad (3)$$

where $x_n$ is the hydrophone coordinate, the origin of which is at the subantenna center. The phasing directions are chosen in increments of 30 degrees: $\theta = -90°, -60°, \ldots 90°$. A directional pattern with suppressed side lobes is employed. The energy directional patterns for a 2020 Hz signal phased in each of the seven directions are shown in Figure 1a. The antenna does not distinguish between left and right directions. A possible improvement to eliminate this problem is the use of vector sensors, as in [19].

When the phasing procedure is completed, the reverberation arrival time is determined by the threshold value (Figure 2). After the reverberation is chosen, its spectrum is calculated in eight windows with a length of 64 samples with an overlap of 75%. Time delays are $\tau = 0; 0.34; 0.675; \ldots 2.7$ s. The phasing and time strobing grid are shown on the map in Figure 1b. The windows for the minimum and maximum delays are shown in Figure 2 in red and magenta, respectively.

In our previous study, we used a regular spectrogram computing routine [16]. Since transmitting and receiving signals were not synchronized, we experienced some jitter in signal arrival times relative to the spectrogram frames. Thus, an additional delay was introduced to ensure that the direct arrival would be skipped. This time, we detected the beginning of reverberation at the threshold, which allowed for more accurate selection of the placement of the Fourier window. This enables processing of the received signal with minimum delay and and elimination of the blind zone near the receiver.

The reverberation spectra should be averaged by the pulse number for better synchronization and more reliable spectrum characteristics. We averaged over four pulses corresponding to a 6 min interval. The ADCP, which is used to obtain benchmark data, provides current data for 1 min. Analysis of these data shows the presence of a long-term up/down trend (which is of interest for measurement) and small fluctuations on a typical time scale of 3–6 min, which are considered noise. Thus, we calculated the averaged ADCP output in the same way as for our own data, namely, the current was also averaged with a 6 min interval.

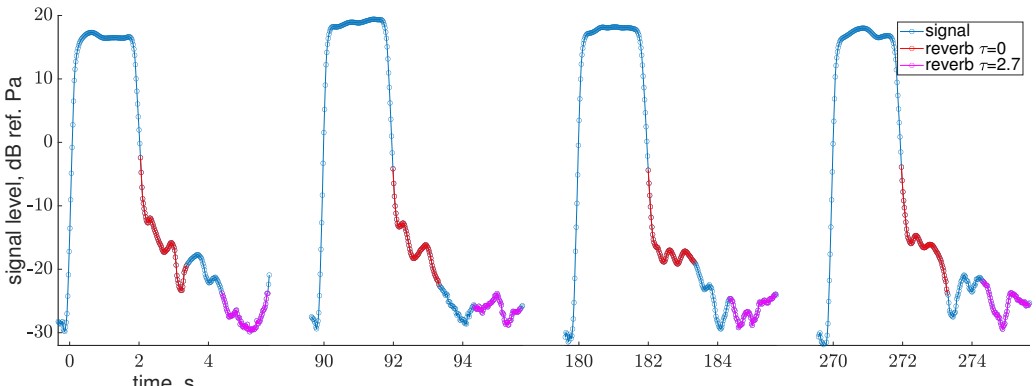

**Figure 2.** Time dependence of a $\theta = -30°$ phased and heterodyned signal on a logarithmic scale with a carrier frequency of 2020 Hz and a reverberation start threshold of 0 dB. Four pulses of the reverberation spectra are averaged in Figure 3b.

Going back to the studied acoustic signals, we should note the following problem. Averaging is carried out with the exception of noisy signals. A narrow-band signal is emitted, while noise interference usually has a wide band. The characteristic Doppler shifts caused by propagation do not exceed 2.5 Hz. The exclusion criterion is as follows: the reverberation spectrum received with a minimum delay is considered, the maximum is in

the range of $[-2.5; 2.5]$ Hz, and the number of points exceeding the maximum minus 9 dB outside the range of $[-2.5; 2.5]$ Hz is considered. If such points account for more than 2%, then the signal is considered poor. An example of such a "poor" pulse is shown in Figure 3a.

According to the averaged spectrum for each direction and time delay, the spectrum features, which simply referred to as features below, are calculated. Following the approach described in [20], we calculated the following values and corresponding notations:

- sk_le and sk_ri are the left and right slopes in a given interval $\pm[0.75, 1.75]$ Hz, respectively;
- centr1 and centr2 are the weighted and median average frequencies, respectively;
- sk_le2 and sk_ri2 are the spectrum slope in a more distant interval ($\pm[1.5, 3.2]$ Hz for $f_0 = 2020$ Hz);
- lvl is the average signal level in the analysis window;
- lv_bragg_le and lv_bragg_ri are the signal level in the 1 Hz band around the Bragg frequency ($\pm\sqrt{gf_0/\pi c} = \pm 2$ Hz if $f_0 = 2020$ Hz).

Figure 3b shows how the spectrum is described in terms of the above features.

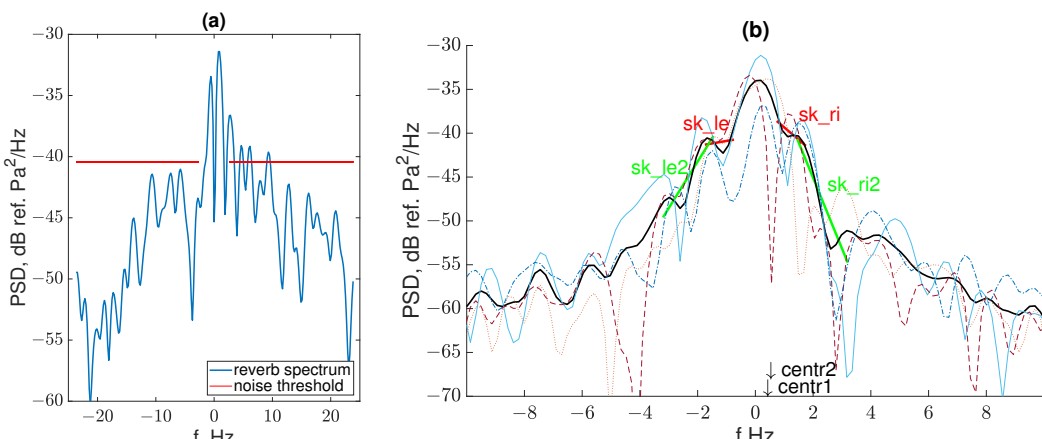

**Figure 3.** Examples of backscattering spectra: (**a**) a noisy reverberation spectrum; (**b**) spectra that passed the quality check (with some annotations). Thin lines are the reverberation spectra with the minimum delay ($\tau = 0$ s) of four pulses shown in Figure 2, the black thick line is the result of averaging, colored straight lines are the spectrum features (slopes), and arrows are mean frequencies.

## 4. Water Flow Parameter Estimations with Machine Learning

Flow parameter estimation can be considered by developing a function ($f : X \rightarrow R^2$) that maps an input signal ($x \in X$) to a flow vector ($v \in R^2$). Because the codomain of $f$ is $R^2$, we build $f$ as a tuple ($(f_{\mathbf{x}}, f_{\mathbf{y}})$) of functions $f_{\mathbf{x}} : X \rightarrow R$ and $f_{\mathbf{y}} : X \rightarrow R$ independently. According to $f_{\mathbf{x}}$, we denote the east projection of the water flow vector, and according to $f_{\mathbf{y}}$, we denote the north projection. In this work, we construct functions $f_{\mathbf{x}}$ as a composition $h_{\mathbf{x}} \circ g$ and $f_{\mathbf{y}}$ as $h_{\mathbf{y}} \circ g$, where $g : X \rightarrow R^{504}$ is a function that maps an input signal vector ($x \in X$) to a feature vector ($\tau \in R^{504}$). $h_{\mathbf{x}}$ and $h_{\mathbf{y}}$ are the functions that are constructed by a machine learning algorithm at the training stage. In the present work, we used the extreme gradient boosting algorithm as the machine learning algorithm [21]. The XGBoost algorithm operates through an iterative process, constructing a sequence of decision tree models, each of which focuses on the residual errors of the previous model to successively refine its predictions. The description of the algorithm is too extended to include here. An example of a decision tree is plotted in Figure 4. The decision tree outputs (i.e., leaf values) are summed up to obtain $h_{\mathbf{x}}$ and $h_{\mathbf{y}}$.

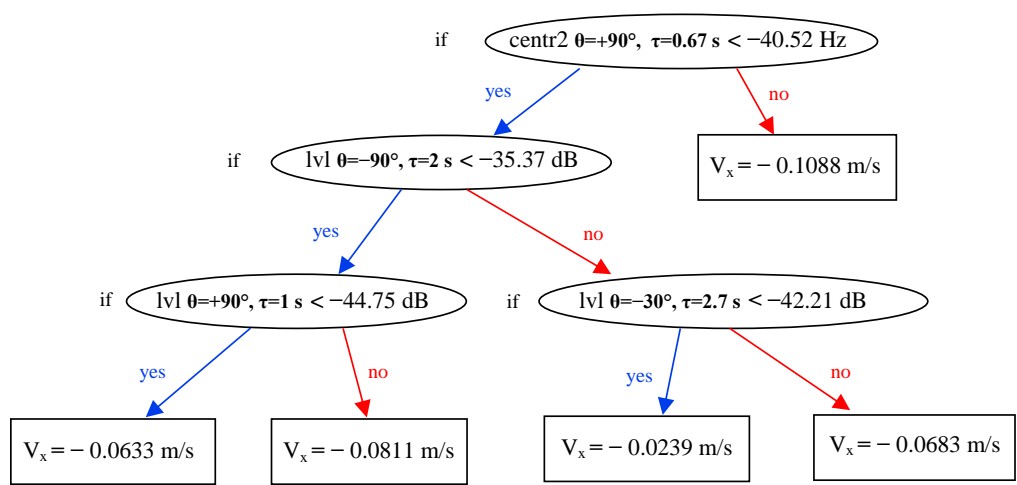

**Figure 4.** An example of a decision tree used to compute the x component of the current velocity.

We build a function (*g*) as a concatenation of all features listed above: sk_le, sk_ri, sk_le2, sk_ri2, centr1, centr2, lvl, lv_bragg_le, and lv_bragg_ri. Every feature was obtained for all delays up to the point when SNR was low and from all directions. In our setting, the number of directions was seven, the number of features was nine, and the number of considered delays was eight; therefore the dimensions of feature vector $\tau$ were $7 \times 9 \times 8 = 504$.

The ADCP data averaged over 6 min were taken as the true current values. In total, including breaks, there were 9 days when the ADCP and underwater acoustic system operated simultaneously. The total number of points was about 1200 measurements. This sample was split into a training and a test sample at a ratio of 10:1.

As a result of the training, a decision tree was constructed. The table of feature importance for a 2020 Hz carrier frequency for a model with 200 trees is presented in Figure 5. Each number next to the column indicates how many times the feature was used to split the data across all trees. Finally, Figure 6 shows the results of predictions on the test set.

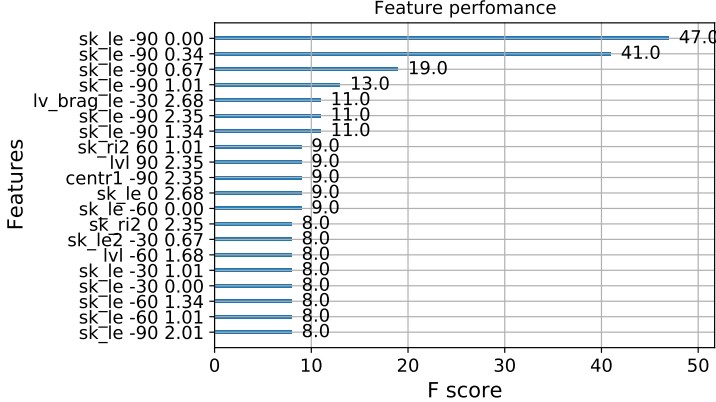

**Figure 5.** Top 20 feature importance for a 2020 Hz carrier frequency. "F score" value indicates how many times the feature was used to split the data across all trees. In total, the model comprises 200 trees.

The accuracy of the predictions estimated by MAPE (mean absolute percentage error), which was calculated according to Equation (4), is given in Table 1.

$$MAPE(F, A) = \frac{1}{N} \sum_{i=1}^{N} \left| \frac{F_i - A_i}{A_i} \right| \qquad (4)$$

where *F* is the prediction, and *A* is the true value. When calculating the error, terms with small values in the denominator ($A_i < 0.01$ m/s) were excluded.

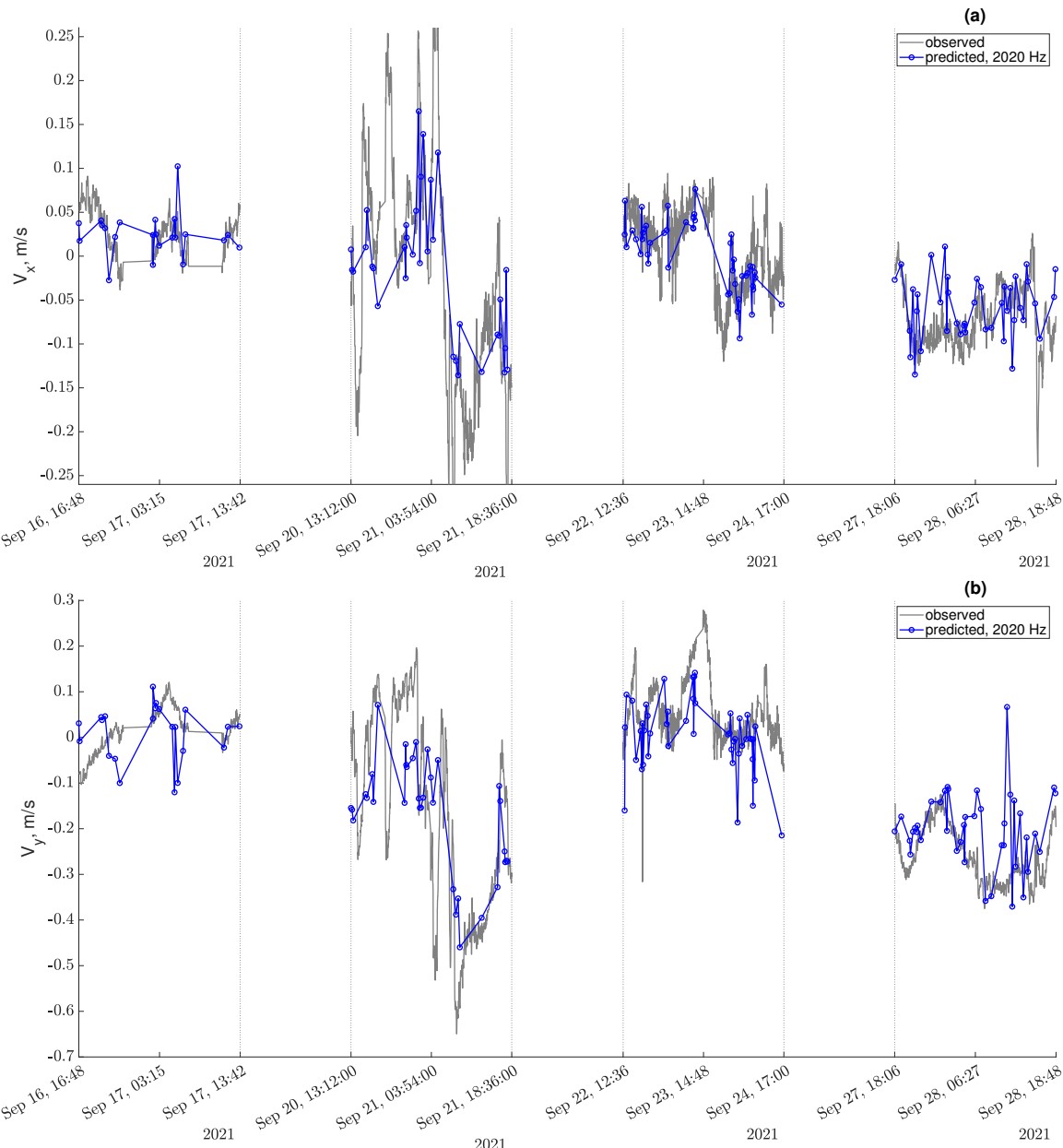

**Figure 6.** Blue circles are the predictions of the XGBoost regression model for a test sample with a carrier frequency of 2020 Hz. The gray solid line is the true current values measured using ADCP indicated at all points where the underwater acoustics and flow meter operated simultaneously. (**a**) Top panel: x component of the current; (**b**) bottom panel: y component of the current.

**Table 1.** Prediction accuracy.

| $f_0$, Hz | MAPE $V_x$, % | MAPE $V_y$, % |
|---|---|---|
| 1320 | 81.2 | 104.0 |
| 2020 | 73.7 | 117.7 |
| 2720 | 99.0 | 122.0 |
| 2080 | 88.3 | 123.0 |

Each feature is associated with a cell on the map, as shown in Figure 1. A feature is related to an angle indicating the direction of the echo signal origin and a delay that



can be easily converted to a distance. In theory, if the hydrological parameters in the surrounding area are similar, other features can be used to obtain current values in other points by passing them through the computed decision tree. Translation and rotation of the coordinate system can be used to select such new features from the set. However, this option has not been fully tested yet. In the current experimental design, the ADCP providing reference values was located in the blind zone of our acoustic system, making the proposed translation of the coordinate system invalid. Nonetheless, it is worth noting the potential for future use.

The proposed method of measuring water currents can be summarized as follows:

1. Transmit sound pulses with a frequency of 2 kHz, a duration of 2 s, and a period of 90 s (with a 50% allowable change in values);
2. Receive echo signals on a horizontal line array that provides a resolution of at least 30 degrees;
3. Compute the complex envelope defined in Equation (2) and the phased signal according to Equation (3);
4. Use signal gating to extract the patch of early reverberation, as highlighted by the red curve in Figure 2, and several subsequent patches;
5. Compute spectra of the extracted patches for each angle of phasing, as shown in Figure 3;
6. Compute the values of the features listed in Section 3.

The remaining processing steps depend on the stage of the experiment. During the training stage:

7. Obtain reference values of water current at 6 min intervals;
8. Pass feature values and reference values into the XGBoost algorithm to determine the rules for computing water current projections in the form of a sum of decision trees. The algorithm generates a set of decision trees consisting of a series of if...else conditions that are linked together. Each condition compares a single feature value with its threshold, and the algorithm computes which feature is used in a node and its corresponding threshold;
9. Extrapolate the decision tree to assess the surrounding water area.

During the testing stage:

7a. Pass feature values and reference values though the decision trees to obtain current projection values. This processor follows a series of if...else conditions, comparing a single feature value with a threshold each time. Depending on which conditions are satisfied, the processor assigns a value to the desired current projection.

In the current experimental design, step 9 could not be fulfilled due to the reason explained above. Therefore, we split the data points over time into training and testing sets to evaluate the proposed method. The results of computing step 7a for testing of data points are shown in Figure 6, and the error is provided in Table 1.

## 5. Correlation Analysis

Our proposed method failed to estimate water current velocity with an error of less than an order of magnitude. Thus, the collected data merit more thorough study and manual checking. According to a primitive model of conditional scatter motion, together with the flow in a boundless water volume, one can associate the flow velocity ($U$) and its direction ($\alpha$) relative to the acoustic line of sight with the scattered signal frequency shift ($\Delta f$) in a well-known way:

$$\Delta f = \frac{2U}{c} \cos \alpha f_0 \tag{5}$$

We relate the value of $\Delta f$ to the feature denoted as "centr1" or the Doppler centroid. In the primary analysis, a direct comparison between $\Delta f$ and $U$ was made. To ensure the most accurate comparison with the $U$ and $\alpha$ values measured by ADCP, we selected the smallest possible signal-gating delay to calculate $\Delta f$. In Figure 7a, one of the flow components and

the Doppler centroid (along the least noisy direction) are plotted on the same axes. For this figure, the two most successful time intervals were used. Specifically, for the Figure 7a, we used the results of the processing performed in [16,20], which differ in a coarser time scale from those reported in the rest of the paper. The Doppler shift was calculated from the antenna directional channel ($\theta = -60°$), where two beams in the horizontal plane take place. The angles between the beams and the inverted selected flow component are $\alpha_1 = 0°$ and $\alpha_2 = 60°$. Due to difficulties in determining the angle between the beam and the flow (caused by the ambiguity), the diagram shows the same type of Doppler function of the centroid depending only on the modulus of the flow current (see Figure 7b).

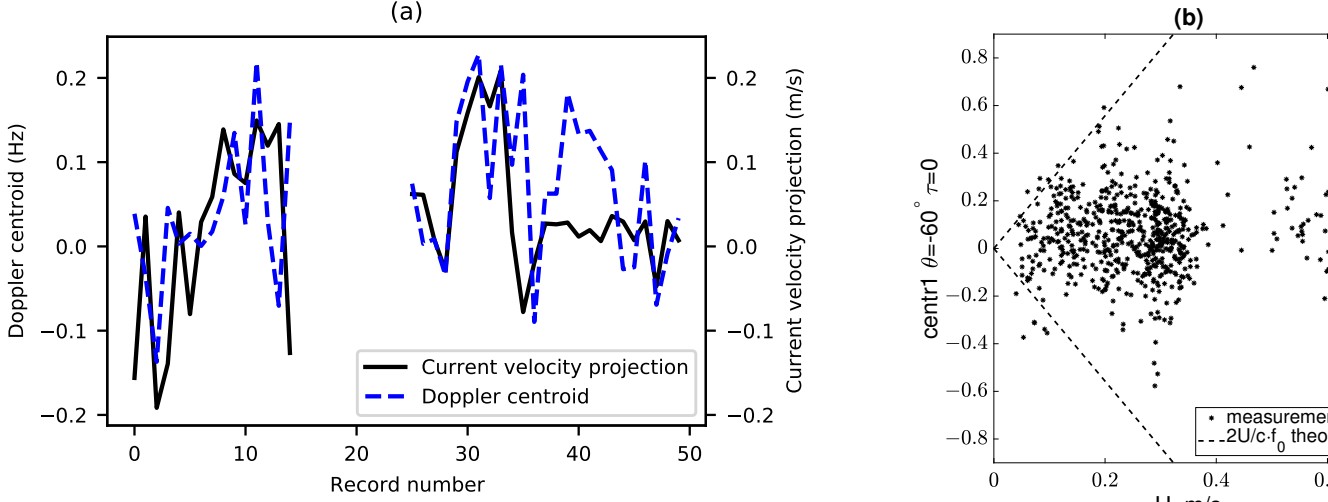

**Figure 7.** Dependence between the Doppler centroid (centr1) feature of the spectrum and the real flow parameters: (**a**) wave features averaging over 20 pulses; (**b**) current features averaging over 4 pulses. The angle of $\theta = -60°$ and minimal delay were used to calculate the acoustic spectrum from which the centroid was constructed.

Figure 7 shows that the temporal variations of the flow and the Doppler centroid of the scattered signal at 2 kHz correlate with each other. However, the coefficient of proportionality is less than the expected value of $2f_0/c$. One may note some cast-off values in the diagram. During processing, we noticed that some observation conditions could be considered favorable, and some could not. Sound propagation in a shallow sea over considerable distances leads to distortion of the Doppler spectrum of the scattered signal. The associated scattering on stationary objects is a deleterious effect. The abovementioned bifurcation of the directional pattern, which is typical of linear phased antenna arrays, is indeed harmful, since the occurrence of a lateral current (across the hydrophone array) does not shift the Doppler centroid but leads to the spectrum broadening in the corresponding directional channel. Therefore, taking into account all possible factors and, at the same time, adjusting for local measurement conditions, we decided to apply machine learning methods in the previous section.

One more thing was left unclear after the formal analysis: whether we can consider data (e.g., Doppler centroids) obtained with longer time delays as a tool for estimating current velocities at remote points. The value $\overline{\Delta F_d^2}$ represents different Doppler centroids obtained at different resolution cells along a single line of sight ($\theta$):

$$\overline{\Delta F_d^2} = \frac{1}{I_d} \sum_i (\Delta f_{d,i}(\theta, \tau_1) - \Delta f_{d,i}(\theta, \tau_2))^2 \qquad (6)$$

where $\Delta f_{d,i}$ is the centr1 value obtained according to the $(\theta, \tau_2)$ direction and the delay in the $(d, i)$ record, while index $d$ is the day, $i$ is a data point on day $d$, and $I_d$ is the total

number of data points available on that day. Averaging over index *i* was performed for all data recorded on day *d*. $\tau_1$ and $\tau_2$ are two different time delays that correspond to different ranges at which Doppler centroids are measured.

The value of $\overline{\Delta F_d^2}$ was used to sort the data, namely to identify samples that were worth paying special attention to, e.g., a large value was found on 21 September 2021 (see Figure 8a). A set of normalized Doppler spectra for a half-hour interval within that day was plotted against the distance in Figure 8b. The aim of that figure was to show that a different spectral shape can be observed in the echo signal coming from different points.

Second, one might be interested to detect whether the introduced value $(\overline{\Delta F_d^2})$ really indicates the occurrence of some extraordinary large-scale turbulence in the sea. No extreme weather conditions were registered on the platform during this time. The only peculiar thing that should be mentioned about the day of the experiment is that ADCP showed inhomogeneous flow along the water column. In contrast to other periods, when all water layers tended to move at same speed, several short bursts of upper-layer current were noted on this day. Each burst lasted about one hour. Thus, a hypothesis that some mesoscale perturbations were drifting in the region is true, and Figure 8 may be considered as evidence of these phenomena. To date, we have not been able to truly validate this result, since we had no control at remote points.

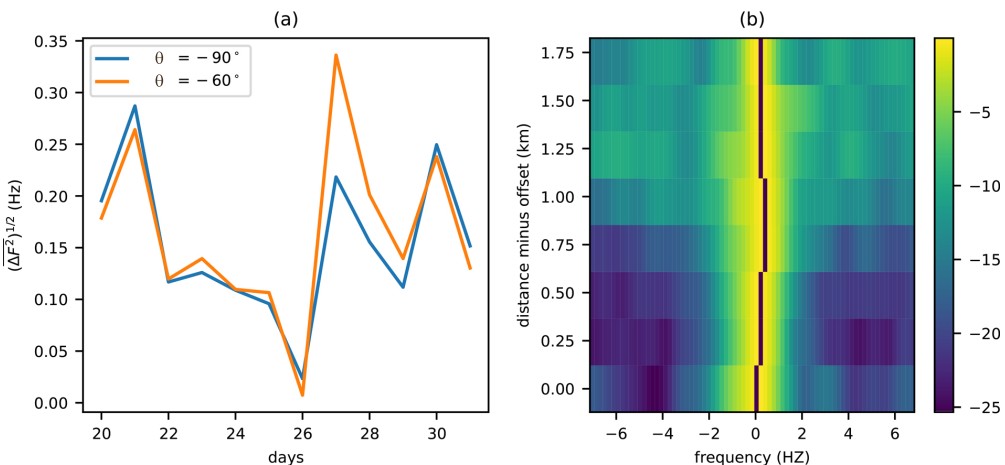

**Figure 8.** Illustration of the use of acoustic remote sensing for tracking of a specific regime of surface currents: (**a**) daily averaged values of the denoted quantity related to the presence of large-scale eddies; (**b**) scattering spectra obtained simultaneously at different distances (dB relative to the maximum value).

## 6. Discussion

Table 1 shows that both velocity projections can be estimated with an error of the order of its magnitude. Thus, the tested acoustic system can be considered a flow indicator on a kilometer scale (recall that Figure 8 proves the potential operating range of the system). The origin of the high values of the mean average percent error is worthy of further discussion.

First, some errors may be conditioned by the comparison method. The predicted values are averaged over an area of about a kilometer, while the true values are point-measured. Some velocity distribution over the area may lead to a mismatch. The occurrence of two lobes on the reception directional pattern can also lead to incorrect averaging.

Second, this is the second course of experimental data processing, which was originally used to study surface waves [16]. Decimeter-scale surface waves are much faster than sea currents. When we deal with currents, a typical Doppler frequency appears to be close to the frequency resolution. Thus, the signal design is not perfect here.

Third, the theory of sound scattering at frequencies of about one kilohertz is not very straightforward. The sound is repeatedly reflected from the surface and bottom as it propagates through the wave guide. It is not clear which kind of agent is responsible for

the transfer of the water column velocity into the backscattered signal shift. Small-scale turbulence has a low scattering strength within this frequency range. Similarly to [13], we expected that the reverberation spectrum would be formed by Bragg scattering on surface waves that correspond to half the sound wavelength. Accordingly, it can be assumed that the scattered signal spectrum will have a discrete form, namely two Bragg peaks.

In contrast, the present paper (e.g., Figure 3) shows that we dealt with smooth spectra. As mentioned in [22], the following phenomena should be taken into account. First, the short waves, which are Bragg reflectors for sound, are modulated by long waves; therefore, the backscattering spectrum is blurred. Secondly, some small scatterers, which are possibly air bubbles, follow the currents inducing the wave. This kind of scattering, together with bottom scattering, creates a wide Gaussian peak in the center of the spectral diagram. The types of scattering listed above, i.e., Bragg waves and bubbles, are maintained according to the law (5) in the presence of a mean water flow, but this is not true for bottom scattering.

Without going deep into theory, we introduced the kinds of features, that, in our opinion, are responsible for different kinds of scattering, e.g., lv_bragg is directly responsible for Bragg scattering, sk_le-sk_ri corresponds to central Gaussian broadening, and sk_le+sk_ri and centr1 are related to the central Gaussian shift (we are still discussing the diagram in Figure 3). Then, we let the machine learning tool create the best combination of these features using the training procedure. As a result, some details could be missing.

Authors may compare the achieved accuracy with what has been provided by the exploited (reference) ADCP. Generically speaking, ADCPs are accurate industrial tools used to measure water currents. In most cases, they are exploited with a vertical orientation option (upward-looking or downward-looking). The ADCP exploited on the platform operated at a frequency of 1.2 MHz was set to a sensing mode with a cell size of 0.25 m, and the entire 30 m water column was probed. The standard deviation of single-ping velocity is estimated as 28 cm/s, while default averaging over 50 pings reduces this error to 4 cm/s [6,23]. ADCP has an internal error control algorithm; thus, under some conditions, some cells are automatically marked with a black flag, showing that such data are unreliable. The most low-frequency representative of that ADCP series operates at a 75 kHz frequency. The greatest sensing range is 600 m, which is achieved when the cell size is set to 32 m. Therefore, the ADCP is still excellent for local measurements . When one needs to obtain a current distribution over a water basin, then ADCP is usually towed by a ship in parallel traverses. Remote sensing tools are more helpful in this case, and the proposed method should be compared with this class of methods.

Researchers who developed the coastal acoustic tomography (CAT) method reported [10] that differences in ADCP results ranged from 0.1 to 1 m/s in various tests, corresponding to 20% to 100%, respectively. Model-based data assimilation helped to improve the accuracy in the least-accurate case. As previously mentioned, CAT requires at least two stable moorings for the instruments. The typical distance between them is 2 km, which should be considered the operational range of such a system. The integral value of speed along the acoustic path is provided by CAT. Sometimes two or three variables can be extracted from a pair of transmitters due to multipath propagation, but in general, the number of transmitters should be increased in order to obtain more independent variables as a result of processing. Meanwhile, the transducer and the receivers are collocated in the proposed method; thus, we need only one mooring point, and a set of speed values is obtained from a single ping due to forward and return sound travel time analysis. The accuracy and the range of CAT operations and the proposed method are of the same order.

Typical parameters of HF radar systems, as provided in [13], are described as follows. One system is equipped with an 83 m long antenna array and provides a range of 15 km and a resolution of 0.25 km. Another system is facilitated with an approximately 870 m long antenna array and provides, under favorable conditions, a range of 200 km and a resolution of 7.5 km. Therefore, HF radars are large facilities suitable for operation from the shore only. The CODAR website reports a velocity resolution of approximately 4 cm/s [14].

Summarizing the mentioned problems of the proposed method and a brief review of the possibilities of other methods, we consider our results as a good experimental demonstration of the possibility of monitoring sea currents using midrange acoustic backscattering. The proposed method is suitable for some practical applications. Finally, the reader might be interested in possible ways of improving the performance of the proposed method. In future studies, we intend to upgrade the acoustical system design. Changing the waveform to a continuous chirp signal will be helpful if accurate resolution at close distances is desired. Perhaps a more advanced theoretical study on acoustic backscattering would result in a more sophisticated signal processing algorithm.

## 7. Conclusions

The experimental results demonstrate the potential of solving the inverse problem of flow prediction based on the spectrum of horizontal backscattering of sound in the middle frequency range. The system tested in this study can provide an estimation of two velocity projections with an error of the order of its magnitude with an operation range of 2 km.

An analysis of potential problems was performed. The initial and rather obvious relation of the water flow velocity with the Doppler centroid of the scattered sound was too simple and too rough, since in practice, the scattering spectrum appeared to be distorted by another kind of scattering, such as bottom reverberation. The predominant flow direction was along the coast, and the x component of the current was predicted with higher accuracy, despite the ambiguity of the system of reception from left to right in this direction. While the ambiguity of the y axis is less pronounced, numerous close-to-zero values for y components of flow velocity were obtained.

Some suggestions to improve the experimental technique were developed. Further experiments measuring points distant from the receiving system and using a well-directed receiver would help to develop an underwater acoustic system for remote monitoring of marine environment parameters from all angles.

**Author Contributions:** Supervision, A.V.E.; project administration, M.B.S.; investigation, A.V.E., I.A.K., D.A.K., D.D.R., and M.B.S.; data processing, D.D.R.; software, A.A.P., and D.D.R.; formal analysis, A.A.P.; writing—original draft, A.A.P., D.D.R., and M.B.S. All authors have read and agreed to the published version of the manuscript.

**Funding:** This research was supported by the Russian Science Foundation (grant number 20-77-10081). The instruments used for underwater acoustic measurements were provided to the team through a State Contract with the Ministry of Education and Science of the Russian Federation (grant number 0030-2021-0017).

**Data Availability Statement:** This research is based on the dataset provided in [20]. The ADCP data that are missing in the set are available upon request.

**Acknowledgments:** The authors are grateful to N.A. Bogatov and A.A. Molkov for their help during the experiment and valuable discussions. Section 4 was prepared within the framework of the Basic Research Program at the National Research University Higher School of Economics (HSE).

**Conflicts of Interest:** The authors declare no conflict of interest.

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
