# Peer review of "Monitoring Sea Currents with Midrange Acoustic Backscattering"

_water, doi:10.3390/w15112016_

Round 1

Reviewer 1 Report

The authors present a method of water flow estimation and evaluate it qualitatively, and quantitatively. While the authors describe a particular application of the method in a use case study fashion, it is not difficult to imaging the broader scope of the work in general. However, this reviewer has concerns regarding the manuscript on the following fronts:

1. The abstract does not seem the well-formed enough to attract attention of potential readers, and thus seemingly affects the impact of the manuscript. A better structuring and ordering of the sentences in the abstract could help, along with a summary of the results obtained.

2. There seems to be a problem with the software used to produce the PDF, as several reference numbers within square brackets have been replaced by question marks.

3. The authors do not seem to devote enough attention to comparing their proposed method with existing ones in a quantitative fashion, or explaining why such a comparison is not possible. For example, while the authors briefly touch upon some methods used to monitor sea currents, they do not seem to present a numeric comparison of their proposed method and those methods, or with potentially other existing ones that have a similar objective as theirs.

Author Response

Dear Reviewer:

Thank you very much for your attention to our paper.

First of all, we are very sorry that the previous manuscript contained no reference list at all. We have definitely fixed this in current edition, as well as we have done improvements, related to the other arisen issues. We provide the list of changes and our comments below. Moreover to that changes and additions are marked with the blue color in the text of the paper, while the deletions are marked as red.

Q1: The abstract does not seem the well-formed enough to attract attention of potential readers, and thus seemingly affects the impact of the manuscript. A better structuring and ordering of the sentences in the abstract could help, along with a summary of the results obtained.

A1: We have revised the abstract

Q2: There seems to be a problem with the software used to produce the PDF, as several reference numbers within square brackets have been replaced by question marks.

A2: Once again we are very sorry about that. These manuscript layout errors have now been corrected. (It turns out that on cannot just upload a zip archive from Overleaf to MDPI)

Q3: The authors do not seem to devote enough attention to comparing their proposed method with existing ones in a quantitative fashion, or explaining why such a comparison is not possible. For example, while the authors briefly touch upon some methods used to monitor sea currents, they do not seem to present a numeric comparison of their proposed method and those methods, or with potentially other existing ones that have a similar objective as theirs.

A3: We have added the requested comparison to the discussion section. See lines 346-380. The precision and the operations range of the coastal acoustic tomography and the proposed method is of the same order. Typical ADCP and HF radar over performs the proposed device. However, ADCP is designed for local measurements, i.e. in points, and HF radars are large facilities suitable for operation from shores only. So the proposed method is definitely going to be suitable for some practical applications

Finally, the authors are very grateful to the reviewer for the valuable comments.

Reviewer 2 Report

The work presented looks good, I like the results and the methodology followed. But I think it is necessary to fix some parts of the article and be more specific in others to bring out its full potential. Here are a few specific comments:

- I think the abstract is not well written. "For a research paper, an abstract typically answers these questions: PURPOSE: What is the nature of your topic/study and why did you do it? METHODS: What did you do, and how? RESULTS: What were your most important findings?". In your abstract, this information is not appear. Please, repeat it.  On the other hand, try not to mix up the units used, it talks about "kilometers" (line 3) and "nautical miles" (line 6). In your abstract you don't talk about "machine learning", however, it appears in the keywords. Please give concordance to the use of keywords related to the abstract and the work in general.

- I consider that the introduction part does not contextualize the work sufficiently. A good introduction should introduce the reader to the subject, generalizing and summarising the work that exists prior to the one presented, then it should explain the idea of what is going to be presented and finally end with a paragraph summarising the content of each of the sections of the present article.

- I have generally noticed a lack of specifics in the text. This is a scientific text, and as such, we should avoid relative expressions. I mean when using "several kilometers" in the abstract, in lines 45, 47-48, 60-61, "the upper layer" in line 17 and 92, "large distances" in line 36, "near the surface" in line 101, "upper and lower layers" in line 103, "low-frequency" in line 105, "thousand oscillation periods" in line 63, "narrow-band" in line 136 etc. Please be cautious and specify at least an approximate range so that the reader can get a better idea of what you are talking about. Maybe some figure schematic might help the reader to clarify some of these concepts. 

- I would also like to read some more detail on some aspects such as: what is the typical accuracy of a commercial ADCP (lines 23-24) or the mentioned in line 101? Can you consider explaining very briefly about CODAR type radars (line 53)? What is the bandwidth used in the mentioned chirp signals of lines 73 and 75? How do you modify slightly the acoustic signal processing (lines 93-94)?

- The use of acronyms is good in this paper, but I suggest the use of HF and LF (line 73 and 105), as well as defining CW (line 78).

- I suggest creating a schematic figure to visually clarify the connections explained in the sentence "The emitted signal is a rectangular pulse with duration of 2 s with a carrier frequency of 1320, 2020, and 2720 Hz and duration of 8 s with a frequency of 2080 Hz" on line 108-109.

- I suggest creating a diagram figure to visually the procedure followed on lines 122-124.

- There are some formatting errors in the text (please re-read the text very carefully as many times as necessary to avoid such errors): Add space after the full stop (line 67), add space between number and unit (lines 107, 108, caption figure 2 "0dB", lines 140, 151 "1Hz", legend in figure 7 "2020Hz", caption figure 6, etc ), complete the reference to the Figure 1 in line 122, remove double brackets in line 257, missing units in \tau (caption on Figure 3), missing label in color scale Figure 5b, orthography error in "a array" (is "an array") on line 18, etc.

- In general, the equations are well explained, but there is always some term missing in the text: "α" in eq. 2, "f_o in eq. 3", "\tau" in the text, etc

- I suggest posting a map of the sensor locations to better visualize the experiment situation (line 97). It could complete Figure 1.

- Figure 1 is missing the scale. And specific distances could be added.

- The text discusses the array of 13 hydrophones (line 112). As this is a scientific text, it should be reproducible, so I suggest you be more specific with the items used: Is this array commercial? What is the distance between hydrophones and the total size of the array? Are they commercial hydrophones (if so, please add brand, model, and manufacturer)? What are their specifications (RVR and directivity)? For what frequency range are they used?  Do they have a flat response and sensitivity in that range?

- Decibels are dimensionless units and are therefore always accompanied by a reference value. Please indicate this in the Y-axis label in Figure 2.

- It is fine to indicate the exact moment when the recording shown in Figure 2 starts in its caption, but it would be better if the time scale in the figure starts at 0 s, to help the reader understand the temporal distance between the pulses shown.

- About Figure 4a, why do they talk about different units (Hz and m/s) if they seem to be exactly the same values? If they are the same values, the comparison could be completed with a figure showing the direct difference between the two measurements. And explain in the text why the difference is greater in Record Number 40.

- Since Figure 6 shows the F score value, why don't you add the equation for its calculation?

- Figure 7 would be missing captions (a) and (b). And to show your results even better, I suggest creating a figure showing the modulus of v in m/s and the direction in degrees of the current.

- If a decimal precision is used in Table 1, add it to all values even if it is zero.

I qualify "Reconsider after major revision" this text because of the fact that the bibliography section is missing and the citation format, which I detail below. It must be corrected.

- The references section is missing

- Fatal error format on the file, because all cites appear as "[?]"

- As in any quality scientific article, citations are placed at the end of the sentence, not in the middle (as in lines 14, 20, 37, etc). Quotations are used to show the origin that proves the stated claim.

- Missing references in statements "... since it is known for relatively weak sound attenuation in water." (lines 48-49) and "since the resonant length of the surface wave is 36 cm for sound with a frequency of 2 kHz." (lines 59-60). If they are not included, I will assume that these are fruitful results.

- Referring to past works without explaining them in the text gives the impression of wanting to force the reader to read the articles mentioned in order to follow the thread of the text (examples in lines 37, 74, 88, 120, 125, 145, etc.). This is a fatal error. The citations are used to prove a sentence, not to "send more reading work to the reader".

Author Response

Dear Reviewer:

Thank you very much for your attention to our paper.

We provide the list of changes and our comments in the attached response file.

Reviewer 3 Report

The submitted paper presents some experiments dedicated to measure velocity by backscatter acoustic technic using Doppler effect.

Unfortunately, the lack of details and sketch of the configuration, the lack of technique presentation prevent from perfect understanding.

Moreover, no reference is given. A lot of details do not appear as scientific : for example, the signal level in figure 2 (raw signal from sensor I assume) are displayed in deciBel scale without reference. An acoustic sensor measures pressure in Pascal. Typical refence in water acoustic is 1µPa that provides level really higher than 0dB. The scal of Figure 3 is dB (at Hertz ?!). This is noticed PSD (Power Spectrum Density I assume). The unity is not 'at Hertz' but Pa²/Hz or Pa/sqrt(Hz) eventually).

For all these reasons, the actual form of the article could not be published.

The quality of English seems OK. I could understand most of the text even if a particular attention has to be paid for structural and grammar (somme sentences are not completed).

Author Response

Dear Reviewer:

Thank you very much for your attention to our paper for your valuable comments!

We have improved the description of the experiment, equipment and signal processing. See the passages, highlighted with the blue and red color in the new manuscript.

Axes on the graphs are corrected in accordance with the comments. E.g. reference unit in Fig. 2 is 1 Pa, so dB values can be below the zero level.

References bug fixed. That was our fault. Thank you for agreeing to review the manuscript in such an ugly format!

Best regards,

Dmitry Razumov & Mikhail Salin on behalf of the coauthors

Round 2

Reviewer 1 Report

The authors seem to have made substantial changes to their manuscript to address the previous round of review comments, and thus this revised manuscript may be accepted for publication.

Author Response

Dear Reviewer:

We are glad that you have found our previous changes relevant. This time we have made some more changes in the manuscript according to the other Reviewers’ comments. Hope those changes have no negative effect.

We provide the results of the current revision in the following manner. A clean pdf without extra markup is going to be uploaded as the main file in this submission. A diff pdf with blue and red highlighting is attached to this cover letter.

Thank you very much for your attention to our paper!

Dmitry Razumov & Mikhail Salin on behalf of the coauthors

Reviewer 2 Report

· What is the "upper layer of the water area" [line 39] for you? Is it the same as the "upper-layer (1-2 m depth)" of line 130? If it is, you can provide this information only the first time when appears the term, if not, specify every time the depth layer limit.

· What are you want to mean in the sentence "However, these methods might be expensive in large-scale problems."? You should explain it by enumerating some examples. [Introduction part]

· I suggest rewriting the sentence "For example, works [9–11 ] describe a network of receivers and sources for acoustic tomography, which operate at a frequency of about 5 kHz." as "For example, there are some works describing a network of receivers and sources for acoustic tomography, which operate at a frequency of about 5 kHz [9–11 ]." to improve the citation format.

· In general, I prefer scientific papers written in an impersonal way. I know that is not mandatory in this journal, but in my opinion, increases the seriousness of the article. This comment is because I read a lot of "we" in the text.

· In my opinion, this article can not miss a reference to similar previous work as "Liu, Yonggang, Robert H. Weisberg, and Clifford R. Merz. "Assessment of CODAR SeaSonde and WERA HF radars in mapping surface currents on the West Florida Shelf." Journal of Atmospheric and Oceanic Technology 31, no. 6 (2014): 1363-1382. DOI: 10.1175/JTECH-D-13-00107.1". I suggest commenting if your conclusions agree or not with other similar works.

· Please, correct the acronym format: "Continuous wave (CW)" should be "Continuous Wave (CW)" [all acronym letters in the capital]

· You are using "HF" acronym two times and is not defined

· I totally disagree with this sentence: "Current paper follows [15]". I suggest something similar to "This work uses low-frequency range, and it is based on work using mid-frequency range for the same proposes [15]. In this previous work...".

· Write something such as "Work [ 13 ] reports..." or "In the preceding paper [15]..." seems as if the lector is forced to read the bibliography section to understand and know what are referring to. Please, review this citation format and improve the quality of the article readerly.

I think there is good work behind this article, but I still think that it would be more useful to add the figures that I commented on in my previous review:
- I suggest posting a map of the sensor locations to better visualize the experiment situation (line 97). It could complete Figure 1.
- About Figure 7 results: To show your results even better, I suggest adding a figure showing the modulus of v in m/s and the direction in degrees of the current. Also, add a figure to show the difference between both (observed-predicted) to evaluate visually these differences.

In general, I notice some flaws in the citation format (I have commented on some examples). The bibliography, and the citation format, it is a formality that increases a lot the seriousness of your article, review it carefully.

Author Response

Dear Reviewer:

We are glad that you have found our previous changes relevant.

Q1:· What is the "upper layer of the water area" [line 39] for you? Is it the same as the "upper-layer (1-2 m depth)" of line 130? If it is, you can provide this information only the first time when appears the term, if not, specify every time the depth layer limit.

A1: Yes, upper-layer is 2m depth for us, we provide this information in the abstract first, and remind it in the text.

Q2:· What are you want to mean in the sentence "However, these methods might be expensive in large-scale problems."? You should explain it by enumerating some examples. [Introduction part]

A2: This means that it is very expensive to replace remote sensing with thousands of sensors like ADCP. We are not aware of works where very large number  sensors were used in a large-scale problem.  This is a fairly obvious common phrase, we left it unchanged.

Q3:· I suggest rewriting the sentence "For example, works [9–11 ] describe a network of receivers and sources for acoustic tomography, which operate at a frequency of about 5 kHz." as "For example, there are some works describing a network of receivers and sources for acoustic tomography, which operate at a frequency of about 5 kHz [9–11 ]." to improve the citation format.

A3: We accepted the suggestion and used this sentence.

Q4:· In general, I prefer scientific papers written in an impersonal way. I know that is not mandatory in this journal, but in my opinion, increases the seriousness of the article. This comment is because I read a lot of "we" in the text.

We agree. The number of «We» in the text is reduced.

At the same time we have studied scientific papers, authored by native English speakers and have found that they do use “We”. The use of “We” is acceptable when it helps to make a sentence shorter and more understandable than it would be in case of the passive voice or “One..”. So “We” is still kept in some places of our manuscript.

Q5:· In my opinion, this article can not miss a reference to similar previous work as "Liu, Yonggang, Robert H. Weisberg, and Clifford R. Merz. "Assessment of CODAR SeaSonde and WERA HF radars in mapping surface currents on the West Florida Shelf." Journal of Atmospheric and Oceanic Technology 31, no. 6 (2014): 1363-1382. DOI: 10.1175/JTECH-D-13-00107.1". I suggest commenting if your conclusions agree or not with other similar works.

A5: We have cited the work. Thank for pointing on that. The conclusions already made in the paper agree with this particular work in terms of parameters of CODAR and other HF radar performance.

Q6:  Please, correct the acronym format: "Continuous wave (CW)" should be "Continuous Wave (CW)" [all acronym letters in the capital]

A6: Ok, done.

Q7:·You are using "HF" acronym two times and is not defined

A7: Acronym was added at line 107

Q8:· I totally disagree with this sentence: "Current paper follows [15]". I suggest something similar to "This work uses low-frequency range, and it is based on work using mid-frequency range for the same proposes [15]. In this previous work...".

A8: Now it is «In the previous research we constructed a machine learning model to predict surface wave height, period, and wave vector based on midrange sound backscatter spectrum features [16] »

Q9: Write something such as "Work [ 13 ] reports..." or "In the preceding paper [15]..." seems as if the lector is forced to read the bibliography section to understand and know what are referring to. Please, review this citation format and improve the quality of the article readerly.

A9: We tried to put citations at the end of the sentences and clarify what exactly we are citing. Now they are:

«For example, there are some works describing a network of receivers and sources for acoustic tomography, which operate at a frequency of about 5 kHz [9-11] » .
«CODAR website reports a velocity resolution of approximately 4 cm/s [14].»

Q10: I think there is good work behind this article, but I still think that it would be more useful to add the figures that I commented on in my previous review:

- I suggest posting a map of the sensor locations to better visualize the experiment situation (line 97). It could complete Figure 1.

A10: We have previously added Figure 1b with the coastline, phasing directions, an example of the scatter region, distances and sensor locations. The use of third-party maps may violate copyrights, so we drew that sketch.

Q11: - About Figure 7 results: To show your results even better, I suggest adding a figure showing the modulus of v in m/s and the direction in degrees of the current. Also, add a figure to show the difference between both (observed-predicted) to evaluate visually these differences.

A11: A piece of practical outcome of this paper is that independent computation of Vx and Vy through the proposed routine is much more preferable.  We can use for training and predict V as a modulus and even tried it, but the result (accuracy) turned out to be worse than for a separate projection. Angle is a really poor parameter to estimate accuracy since it turns to be inaccurate when the modulus is low. Finally, we could have calculated V = sqrt(Vx^2+Vy^2) and the angle via arctangent, however we decided not to add two more plot since the paper is already large and time for revision is short.

Q12: In general, I notice some flaws in the citation format (I have commented on some examples). The bibliography, and the citation format, it is a formality that increases a lot the seriousness of your article, review it carefully.

A12: As for the above comment: we tried to fix that issue as much as we could. As for the list of the references in the end of the paper: it is generated by LaTeX in accordance with the publisher’s style file.

Besides that we have made some more changes in the manuscript according to the other Reviewers’ comments:

We swapped the places of Sections 4 and 5 to preserve more correct logical order.

A summary of the proposed method is added in the end of Section 4.

Figure 4 was added to show an example of a decision tree.

Small editions were made throughout all the text to make some statements more clear and additional language check was performed.

(Hope the listed changes have no negative effect.)

We provide the results of the current revision in the following manner. A clean pdf without extra markup is going to be uploaded as the main file in this submission. A diff pdf with blue and red highlighting is attached to this cover letter.

Thank you very much for your attention to our paper!

Dmitry Razumov & Mikhail Salin on behalf of the coauthors

Reviewer 3 Report

Main remarks have been taken into account to really improve the article.

The article is worth of publishing even if some details could be still improved: the natural reference in acoustic is not the Pascal but the microPascal. It is not wrong to use another reference but most of people have in mind values scaled with µPa (that makes a gap of 120 dB). etc.

It is still hard to understand some steps of the methods becausefew details are given (equations, example of signals). It has been improved but it remains hard to fully understand the global method.

Generally, it should be better to submit at least the final article and not only the article with modifications.

Author Response

Dear Reviewer:

We are glad that you have found our previous changes relevant. Thank you for the attention to our paper and for your positive evaluation of the first revision.

We tried to do our best to address your general remark: «it remains hard to fully understand the global method». In order to make the method more understandable and the text more readable we have made the following changes:

We swapped the places of Sections 4 and 5 to preserve more correct logical order.

A summary of the proposed method is added in the end of Section 4.

Figure 4 was added to show an example of a decision tree.

Small editions were made throughout all the text to make some statements more clear and additional language check was performed.

We provide the results of the current revision in the following manner. A clean pdf without extra markup is going to be uploaded as the main file in this submission. A diff pdf with blue and red highlighting is attached to this cover letter.

Thank you very much for your attention to our paper!

Dmitry Razumov & Mikhail Salin on behalf of the coauthors

Round 3

Reviewer 2 Report

 I do not agree with your comment "however we decided not to add two more plot since the paper is already large and time for revision is short.". The review time can always be extended if justified, I prefer to publish something (permanently) with as many revisions as needed, we can't publish in a hurry (I know this is not only a question of authors). Even so, you have done a great job throughout the revision process, I hope you will take these comments into account for your next publications.